# A comprehensive biomechanical analysis of the barbell hip thrust

**Adam Brazil**[1]*, **Laurie Needham**[1], **Jac L. Palmer**[2], **Ian N. Bezodis**[2]

**1** Department for Health, University of Bath, Bath, United Kingdom, **2** Cardiff School of Sport and Health Sciences, Cardiff Metropolitan University, Cardiff, United Kingdom

These authors contributed equally to this work.

* A.brazil@bath.ac.uk

## Abstract

Barbell hip thrust exercises have risen in popularity within the biomechanics and strength and conditioning literature over recent years, as a method of developing the hip extensor musculature. Biomechanical analysis of the hip thrust beyond electromyography is yet to be conducted. The aim of this study was therefore to perform the first comprehensive biomechanical analysis the barbell hip thrust. Nineteen resistance trained males performed three repetitions of the barbell hip thrust at 70% one-repetition maximum. Kinematic (250 Hz) and kinetic (1000 Hz) data were used to calculate angle, angular velocity, moment and power data at the ankle, knee, hip and pelvic-trunk joint during the lifting phase. Results highlighted that the hip thrust elicits significantly ($p < 0.05$) greater bilateral extensor demand at the hip joint in comparison with the knee and pelvic-trunk joints, whilst ankle joint kinetics were found to be negligible. Against contemporary belief, hip extensor moments were not found to be consistent throughout the repetition and instead diminished throughout the lifting phase. The current study provides unique insight to joint kinematics and kinetics of the barbell hip thrust, based on a novel approach, that offers a robust evidence base for practitioners to guide exercise selection.

## Introduction

Strength training, known as the process of imposing physical loading to increase the capability of the neuromuscular-skeletal system to produce force, is recognised as an essential component in the physical preparation of athletes to enhance performance [1]. Therefore, a continuous aim of both research and practice is to identify the most effective training strategies to maximise performance enhancements [2, 3].

Recently, there has been an increase in the popularity of the barbell hip thrust, a type of bridging exercise performed against an external barbell resistance, used to develop the hip extensor musculature. Since its introduction to the literature by Contreras et al. [4], the hip thrust has gained popularity within the biomechanics and strength and conditioning communities due to evidence of superior gluteal activation characteristics compared with more conventional resistance training exercises such as the back squat or deadlift variations [5–8].

Due to the horizontally (anterior-posterior) loaded nature of the hip thrust, authors have speculated that this exercise requires a consistent hip extension moment throughout its range

**Competing interests:** The authors have declared that no competing interests exist.

of motion [9, 10], and maximal muscular tension when hip joint reaches full extension [4, 5, 7]. In addition, the loading nature of the hip thrust elicits a horizontal orientation of the resultant ground reaction force vector relative to the athlete in the global coordinate system [11]. This relative orientation is suggested to enhance the transfer of training to athletic performances requiring horizontal force production (e.g. sprinting) based on the "force-vector hypothesis" [10, 12].

Studies have investigated the influence of hip thrust on sprint [10, 13, 14] or jump [10, 11] performance, as well as the relationships between the hip thrust and sprint performance measures [8, 12]. Based on the force-vector hypothesis, the hip thrust should provide a mechanical advantage over traditional standing barbell exercises that elicit a relative vertical orientation of the resultant force vector. Correlational analyses have supported these ideas by evidencing stronger relationships between hip thrust kinetic measures and sprint performance, compared with vertically oriented exercises [8, 12]. However, evidence from training studies is currently equivocal, with Contreras et al. [10] indicating sprint and vertical jump improvements were superior following hip thrust and front squat training interventions, respectively, whereas the hip thrust has recently been found to elicit equal improvements in vertical and horizontal jumping performance [11]. Further, Jarvis et al. [13] observed no significant improvements in sprint performance following a hip thrust training intervention. Differences in study populations, training parameters and performance tests likely provide some explanation for the current lack of consensus regarding the effectiveness of the hip thrust exercise, and signifies the requirement for further training interventions [15], but also comprehensive biomechanical analyses to better understand the mechanisms by which the hip thrust may influence performance [16].

Biomechanical analysis offers insight to the underlying kinematics and kinetics of a training exercise, providing coaches and athletes conceptual understanding to bring objectivity to sport-specific exercise selection [16]. Investigating the musculoskeletal demand placed on the lower limb and pelvic-trunk joints is fundamental to biomechanical analyses of strength training exercises [17–22], although has yet to be undertaken for the hip thrust. Specifically, for the hip joint, whilst authors have proposed that the hip thrust requires a consistent hip extension moment and greater muscular "tension" when hip joint is close to full extension [4, 5, 7, 8], there is currently no joint kinetic evidence to support these ideas.

Therefore, the aim of this study was to perform the first comprehensive biomechanical analysis the barbell hip thrust. Based on the current body of knowledge, it was hypothesised that the hip thrust would, 1) elicit a large hip extensor moment; and 2) hip extensor moments would remain consistent throughout the lifting phase. The purpose of the study was to provide scientists, coaches and athletes with information relating to the external and musculoskeletal demand of performing the barbell hip thrust, to help inform training practices that can positively impact upon athletic performance.

## Materials and methods

Nineteen resistance trained males (age, 22.4 ± 3.1 years; mass, 78.8 ± 11.4 kg; height, 1.77 ± 0.09 m, hip thrust one-repetition maximum [1RM] = 189 ± 42 kg) gave written informed consent to participate in the current study following approval by the Cardiff Metropolitan University Ethics Committee. Participants were free from injury and regularly used the barbell hip thrust in their training routine.

Kinematic data were captured at 250 Hz with a 15 camera Vicon Vantage system (Vicon, Oxford, UK). A marker set comprising 26 individual markers were attached to each participant to facilitate the creation of an eight-segment model (bilateral feet, shanks and thighs,

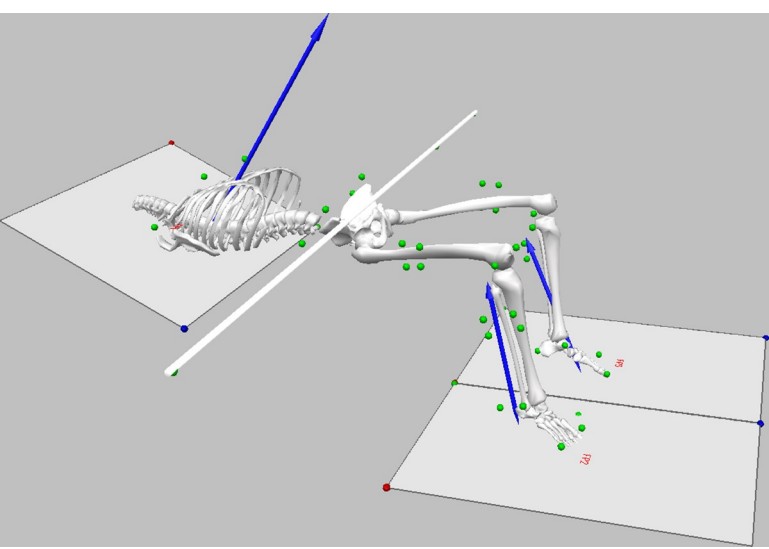

**Fig 1. Image of experimental setup including location of force platforms, direction of resultant force vectors (blue lines), orientation and position of the body segments and barbell, and marker-set (green circles).**

pelvis and thorax), at the following anatomical landmarks: (bilateral), acromion process, iliac crest, posterior superior iliac spine, anterior superior iliac spine, lateral and medal femoral epicondyles, lateral and medial malleoli, first and fifth metatarsal heads, calcaneus, head of the second toe; (unilateral), transverse process of the $7^{th}$ cervical and $10^{th}$ thoracic vertebrae, sternum, and xyphoid process. Rigid clusters comprising four-marker clusters were attached to the lateral aspect of the thigh and shank segments and used for segment tracking, and additional markers were attached to the barbell to track its position and orientation (Fig 1). Synchronised kinetic data were captured using three Kistler 9287CA force plates (Kistler, Winterhur, Switzerland) operating at 1000 Hz. Two force plates were located in standard in-ground dwellings, and were used to measure forces separately at each foot. The third force plate was mounted to a custom-built rig, specifically for measurement of the hip thrust. It was raised above the ground and angled at 20° to the horizontal, to facilitate accurate measurement of external force between the thorax and bench. Following pilot testing, a 15 mm medium density foam mat was secured to the top of the raised force plate to reduce participant discomfort. The rig was positioned such that the participant could comfortably perform the hip thrust with their feet located near the center of the in-ground plates. Participants performed a self-selected warm-up. Data collection comprised one set of three repetitions of the barbell hip thrust at 70% of 1RM determined from recent training data. Participants were asked to perform each repetition with their habitual technique and timing, with maximal intent in the lifting phase towards full hip extension, before controlling the barbell back to the starting position. The barbell came to a rest on the ground before the next repetition was completed, so that each repetition was performed from the same initial starting position (see S1 Appendix for example video trial).

After labelling and gap-filling of marker trajectories (Nexus, v2.6, Vicon, Oxford, UK), data processing was performed using Visual 3D software (v6, C-Motion Inc, Germantown, USA). Raw marker coordinates and force traces were low-pass filtered (4th order Butterworth) with cut-off frequencies of 3 and 30 Hz, respectively, determined through residual analysis [23]. Data from the raised force plate were rotated and resolved into the global coordinate system, defined as a right-handed orthogonal coordinate system of X (medio-lateral pointing right), Y

(anterior-posterior pointing forwards) and Z (superior, pointing upwards). Analysis was undertaken on the lifting (bar-raising) phase of each repetition. The start of the lifting phase was defined when the vertical velocity of the barbell became positive and remained greater than 0 m·s⁻¹. The end of the lifting phase was defined as the point of maximum vertical barbell displacement.

For analysis of external kinetics, Y-Z resultant forces were calculated at the feet (combination of force data for each foot), thorax (force between the thorax and bench) and in total (sum of all forces), of which peak and average values were obtained for the lifting phase and normalised to bodyweight (BW). For joint level analyses, each segment's coordinate system (SCS, defined with the same right-handed orthogonal coordinate system as the global reference) was defined using a static calibration trial. Joint angular velocity was the rate of change of the distal relative to the proximal SCS, described by an XYZ Cardan sequence. Inverse dynamic procedures were applied using Visual 3D software (v6, C-Motion Inc, Germantown, USA) [24] to calculate resultant moments resolved in the proximal SCS at the ankle, knee, hip and pelvic-trunk joints, using the default parameters for segment mass [25] and inertial characteristics [26] (Fig 2). The distal end of the foot was defined at the metatarsophalangeal joint, and the moment acting between the distal end of the foot and the ground was assumed to be zero. Due to the sagittal plane nature of the movement and the extensor demand of the task, x-axis (flexion-extension) data only are reported, with a focus on extensor properties that were defined as positive. Joint extensor (positive) impulse was calculated through integration of the respective moment-time curve using the trapezium rule. Joint power was calculated as the product of joint moment and joint angular velocity. The main phases of positive extensor joint power

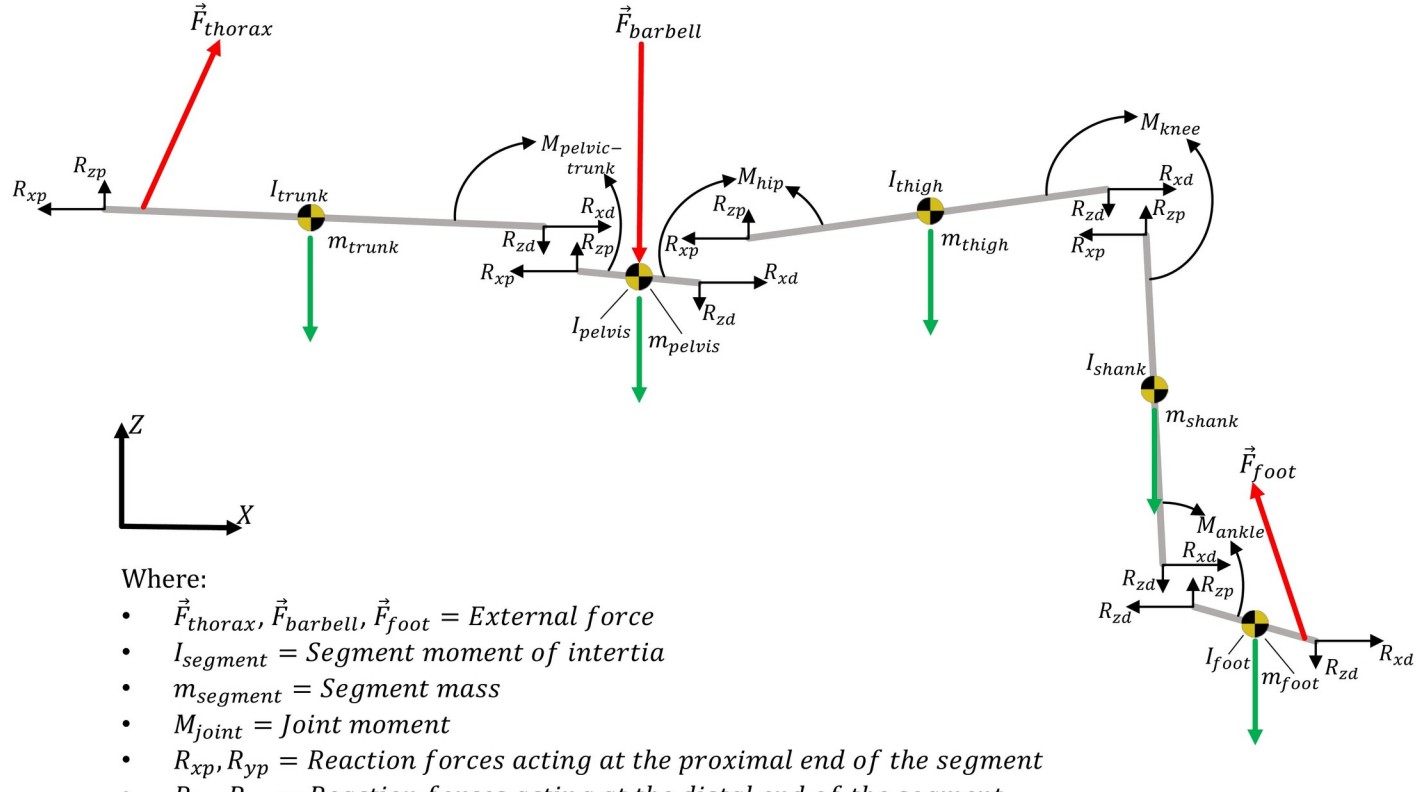

Where:
- $\vec{F}_{thorax}, \vec{F}_{barbell}, \vec{F}_{foot}$ = External force
- $I_{segment}$ = Segment moment of intertia
- $m_{segment}$ = Segment mass
- $M_{joint}$ = Joint moment
- $R_{xp}, R_{yp}$ = Reaction forces acting at the proximal end of the segment
- $R_{xd}, R_{yd}$ = Reaction forces acting at the distal end of the segment

**Fig 2. Representative two-dimensional free-body diagram of the barbell hip thrust.**

were identified, and joint work was calculated for each by integrating power-time curves (trapezium rule), to define positive extensor work performed (energy generation).

Joint kinetic data were normalised to body mass, and extensor characteristics were calculated. Ensemble mean and standard deviations were calculated for all discrete data and for external force, joint kinematic and kinetic time histories, which were time-normalised to 100% of the lifting phase using a cubic spline. All ankle, knee and hip joint data were initially averaged across the two limbs, to present data showing the distribution of loading at individual joints. Subsequently, joint kinetic variables of the hip and knee were each calculated as the sum of the two limbs to provide a valid comparison of musculoskeletal demand with the pelvic-trunk joint. This facilitated greater insight into the total bilateral loading at each joint that was required to raise the barbell.

Unpaired t-tests were used to analyse the differences in external resultant forces acting at the feet and thorax, and a one-way analysis of variance (ANOVA) was used to compare differences in joint kinematic and bilateral kinetic data between the knee, hip and pelvic-trunk joints. When a significant main effect was observed, Bonferroni post-hoc analyses were implemented to identify significant between-joint comparisons. T-tests and ANOVA were conducted using IBM SPSS Statistics (version 26, IBM, Armonk, USA), and statistical significance was accepted at $p < 0.05$. To accompany inferential statistics, Cohen's $d$ effect sizes and 95% confidence intervals were calculated using Estimation Statistics [27]. When confidence intervals of the effect size did not overlap zero, magnitudes of $d$ were interpreted as small ($0.2 \leq d < 0.6$), moderate ($0.6 \leq d < 1.2$), large ($1.2 \leq d < 2.0$), very large ($2.0 \leq d < 4.0$), and extremely large ($d \geq 4.0$) (equivalent scale used for negative values of $d$) [28].

In addition to discrete statistics, statistical parametric mapping (SPM) [29] was used to statistically compare joint moment waveforms between the knee, hip and pelvic-trunk joint. Specifically, a one-way ANOVA with post hoc test was used ($\alpha = 0.05$). Post hoc testing was conducted using SPM independent t-test to provide the scalar output statistic, SPM{t}. Critical thresholds (t*) were adjusted using a Bonferroni procedure. All SPM analyses were done using open source spm1d code (v.04, www.smp1d.org) in Matlab (R2017a, The Mathworks Inc, Natick, USA).

## Results

### External kinematics & kinetics

During the lifting phase, group mean (± SD) vertical barbell displacement was 0.361 ± 0.042 m in a time of 0.828 ± 0.148 s (Table 1). Resultant forces at the feet, thorax and in total are presented in Fig 3. All force curves demonstrated a similar pattern, with an initial peak at approximately 20% of lift time followed by a steady decline and plateau towards the end of the lift. Magnitudes of peak (2.16 ± 0.52 BW vs. 1.69 ± 0.42 BW; $d = 1.00$) and average (1.72 ± 0.36 BW vs. 1.25 ± 0.29 N; $d = 1.59$) resultant force were significantly ($p < 0.05$) greater at the feet compared with the thorax, with moderate and large effect sizes, respectively (Table 1).

### Joint kinematics & kinetics

Continuous joint angle, angular velocity, moment and power data are presented for the ankle, knee, hip and pelvic-trunk joint in Fig 2. The ankle joint exhibited predominantly dorsiflexion throughout the lifting phase, with some individuals transitioning to a period of plantarflexion towards the end the lift. The direction of the net ankle moment demonstrated variation between plantar- and dorsi-flexor dominance, resulting in both positive and negative power to be observed amongst participants (Fig 4). Although variable in direction, magnitudes of ankle joint kinetics were negligible in comparison to other joints (Fig 4).

**Table 1. External characteristics.**

|  | Mean | ± | SD |  | *d* | 95% CI |  |  |
|---|---|---|---|---|---|---|---|---|
| Lift time (s) | 0.828 | ± | 0.148 |  |  |  |  |  |
| Vertical barbell ROM (m) | 0.361 | ± | 0.042 |  |  |  |  |  |
| Peak feet resultant force (BW) | 2.16 | ± | 0.52 |  |  |  |  |  |
| Peak thorax resultant force (BW) | 1.69 | ± | 0.42 | * | 1.00 | 0.64 | to | 1.40 |
| Peak total resultant force (BW) | 3.81 | ± | 0.88 |  |  |  |  |  |
| Average feet resultant force (BW) | 1.72 | ± | 0.36 |  |  |  |  |  |
| Average thorax resultant force (BW) | 1.25 | ± | 0.29 | * | 1.59 | 0.88 | to | 2.35 |
| Average total resultant force (BW) | 2.97 | ± | 0.58 |  |  |  |  |  |

*Denotes significant difference (p < 0.05) between feet and thorax force data. External forces have been normalised to bodyweight (BW).

The knee joint extended for most of the lifting phase, and the hip joint extended throughout the entire lifting phase, which was consistently observed across all participants (Fig 4). Whilst on average there was pelvic-trunk joint extension during the lift, both extension and flexion were observed amongst participants (Fig 4). ANOVA revealed a significant main effect for extensor range of motion, with the hip joint (75 ± 19˚) showing significantly greater range of motion compared with the knee (21 ± 7˚; *d* = 3.80, very large) and pelvic-trunk joint (12 ± 21˚; *d* = 3.21, very large, Table 3).

A net extensor joint moment dominated the lifting phase across the knee, hip and pelvic-trunk joints (Fig 4), with ANOVA revealing a significant main effect for all bilateral joint moment characteristic comparisons (Table 2). A similar moment pattern was observed between the hip and pelvic-trunk joint, with an initial peak at approximately 14% of movement time, before reducing and plateauing towards the end of the lifting phase. Joint moments at

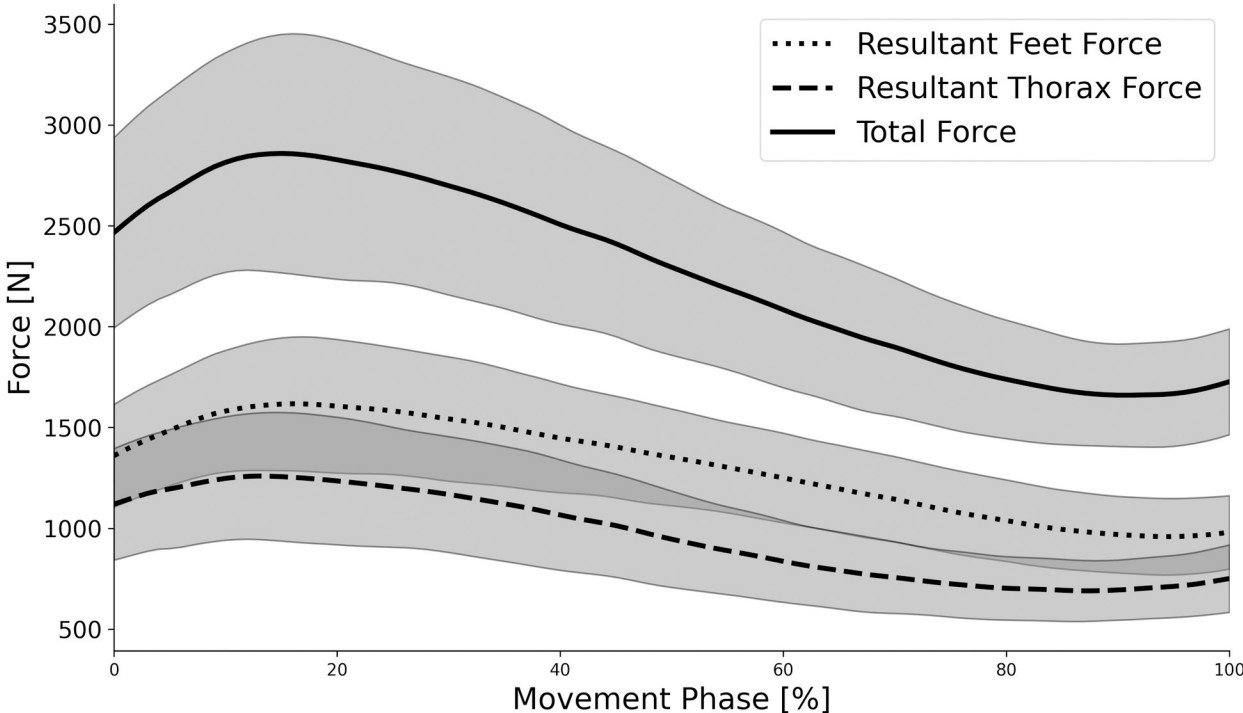

**Fig 3. Ensemble group mean (± SD gray shaded areas) external forces for the thorax-bench (thorax) and foot-ground (feet) interfaces.**

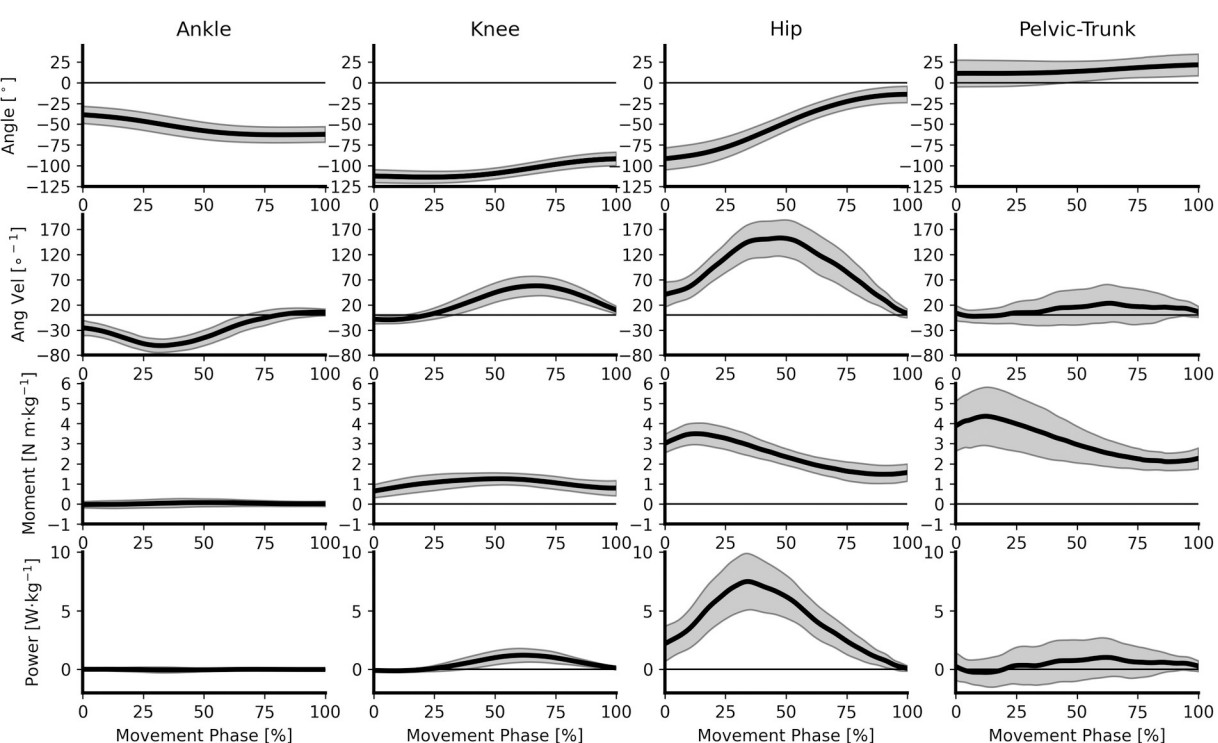

**Fig 4. Group ensemble mean (± SD gray shaded areas) joint kinematic and unilateral kinetic time series data.** Column 1 –ankle joint, column 2 –knee joint, column 3 –hip joint, and column 4 –pelvic-trunk joint. Row 1 –joint angles, row 2 –joint angular velocities, row 3 –joint moments, row 4 –joint powers.

**Table 2. Joint characteristics.**

| | Knee | | | Hip | | | Pelvic-Trunk | | | |
|---|---|---|---|---|---|---|---|---|---|---|
| | **Mean** | **±** | **SD** | **Mean** | **±** | **SD** | **Mean** | **±** | **SD** | |
| *Average of joints* | | | | | | | | | | |
| Extensor range of motion (°) | 21 | ± | 7 | 75 | ± | 19 | 12 | ± | 21 | * |
| Peak extensor moment (N·m·kg⁻¹) | 1.34 | ± | 0.35 | 3.52 | ± | 0.57 | | | | |
| Time of peak extensor moment (%) | 51.0 | ± | 12.6 | 14.3 | ± | 3.1 | 14.3 | ± | 5.3 | * |
| Joint angle at peak extensor moment (°) | -108 | ± | 8 | -83 | ± | 16 | 9 | ± | 18 | * |
| Average extensor moment (N·m·kg⁻¹) | 1.02 | ± | 0.31 | 2.39 | ± | 0.42 | | | | |
| Extensor impulse (N·m·s·kg⁻¹) | 0.82 | ± | 0.25 | 1.97 | ± | 0.45 | | | | |
| Peak extensor power (W·kg⁻¹) | 1.27 | ± | 0.56 | 7.75 | ± | 2.36 | | | | |
| Average extensor power (W·kg⁻¹) | 0.69 | ± | 0.33 | 4.08 | ± | 1.20 | | | | |
| Extensor work (J·kg⁻¹) | 0.43 | ± | 0.18 | 3.23 | ± | 0.88 | | | | |
| *Sum of bilateral joints* | | | | | | | | | | |
| Peak extensor moment (N·m·kg⁻¹) | 2.65 | ± | 0.71 | 6.97 | ± | 1.13 | 4.39 | ± | 1.53 | * |
| Average extensor moment (N·m·kg⁻¹) | 2.03 | ± | 0.42 | 4.79 | ± | 0.84 | 3.06 | ± | 0.90 | * |
| Extensor impulse (N·m·s·kg⁻¹) | 1.64 | ± | 0.50 | 3.95 | ± | 0.90 | 2.46 | ± | 0.72 | * |
| Peak extensor power (W·kg⁻¹) | 2.66 | ± | 1.25 | 15.39 | ± | 4.72 | 2.39 | ± | 1.33 | * |
| Average extensor power (W·kg⁻¹) | 1.37 | ± | 0.67 | 8.16 | ± | 2.42 | 1.18 | ± | 0.72 | * |
| Extensor work (J·kg⁻¹) | 0.85 | ± | 0.36 | 6.47 | ± | 1.76 | 0.74 | ± | 0.59 | * |

*Denotes significant main effect of one-way ANOVA.

**Table 3. Joint characteristics post-hoc comparison effect sizes (*d*) and 95% Lower (L) and Upper (U) confidence intervals.**

| | Hip vs. Knee | | | Pelvic-Trunk vs. Knee | | | Hip vs. Pelvic-Trunk | | |
|---|---|---|---|---|---|---|---|---|---|
| | *d* | L | U | *d* | L | U | *d* | L | U |
| Extensor range of motion (˚) | 3.80 | 2.77 | 4.84 | n/a | | | 3.21 | 2.34 | 4.09 |
| Time of peak extensor moment (%) | -3.98 | -5.32 | -2.84 | -3.78 | -5.14 | -2.72 | n/a | | |
| Joint angle at peak extensor moment (˚) | 2.03 | 1.38 | 2.78 | 8.60 | 6.21 | 11.04 | -5.54 | -6.82 | -3.95 |
| Peak extensor moment (N m·kg$^{-1}$) | 4.58 | 3.38 | 5.73 | 1.46 | 0.58 | 2.38 | 1.91 | 1.17 | 2.55 |
| Average extensor moment (N m·kg$^{-1}$) | 3.75 | 2.69 | 4.67 | 1.34 | 0.54 | 2.08 | 1.98 | 1.20 | 2.67 |
| Extensor impulse (N m·s·kg$^{-1}$) | 3.16 | 2.33 | 3.90 | 1.32 | 0.53 | 2.21 | 1.83 | 1.21 | 2.44 |
| Peak extensor power (W·kg$^{-1}$) | 3.68 | 2.66 | 4.75 | n/a | | | 3.50 | 2.49 | 4.58 |
| Average extensor power (W·kg$^{-1}$) | 3.83 | 2.75 | 5.03 | n/a | | | 3.64 | 2.60 | 4.79 |
| Extensor work (J·kg$^{-1}$) | 4.42 | 3.09 | 5.79 | n/a | | | 4.19 | 2.83 | 5.55 |

All comparisons are significant at p < 0.05 from Bonferroni post-hoc tests from one-way ANOVA. The direction of *d* indicates whether the joint on the left (positive) vs. right (negative) was of larger magnitude.

the knee reached peak magnitudes significantly later in the lifting phase (51.0 ± 12.6%; *d* = 3.98 and 3.78 for hip and pelvic-trunk comparisons, respectively), and again declined towards the end of the lift (Tables 2 and 3; Fig 4). The angle at which peak extensor moments occurred was significantly different between all joint comparisons (*d* = 2.03, very large, to 8.60, extremely large), with the knee (-108 ± 8˚) and hip (-83 ± 16˚) joints in flexed positions, whereas the pelvic-trunk joint (9 ± 18˚) was in a more neutral position. Post-hoc analysis again revealed significant differences between all joint comparisons for the magnitude of peak and average moment, and joint impulse (Table 3). In all cases, the bilateral hip joint elicited the greatest magnitude of extensor kinetics, followed by the pelvic-trunk and knee joints (Table 2). Very large to extremely large effect sizes were observed between the hip and knee joint (*d* = 3.16 to 4.58), with large effect sizes found between the hip and pelvic-trunk (*d* = 1.83 to 1.91), and knee and pelvic-trunk joints (*d* = 1.32 to 1.46) (Table 3). The region of joint moment differences is highlighted in Fig 5, where significant SPM ANOVA main effects were observed. Post-hoc 1D analysis revealed statistically significant joint moment differences, as indicated by supra-threshold clusters, between the hip and knee joint for 100% of lift time, hip and pelvic-trunk joint between 0–78% of lift time, and knee and pelvic-trunk joint from 0–46% and 97–100% of lift time.

Extensor moments were observed at the knee and hip joint as they extended throughout the lifting phase. As such, both joints generated extensor energy as indicated by positive magnitudes of extensor power (Fig 4). Due to inter-individual variability in the direction of pelvic-trunk joint angular velocity, periods of positive (extensor energy generation) and negative (extensor energy absorption) power were shown (Fig 4). Significant ANOVA main effects were again found across all joint extensor power characteristics (Table 2), with post-hoc analyses revealing that the bilateral hip joint elicited the greatest magnitude of peak extensor power (*d* = 3.68 and 3.50, very large), average extensor power (*d* = 3.83 and 3.64, very large), and extensor work (*d* = 4.42 and 4.19), compared with the knee and pelvic-trunk joints, respectively. No significant post-hoc difference was observed between the bilateral knee and trunk for extensor power characteristics (Table 3).

## Discussion

The aim of the current study was to perform the first comprehensive biomechanical analysis of the barbell hip thrust, quantifying the external and musculoskeletal demand of this exercise to

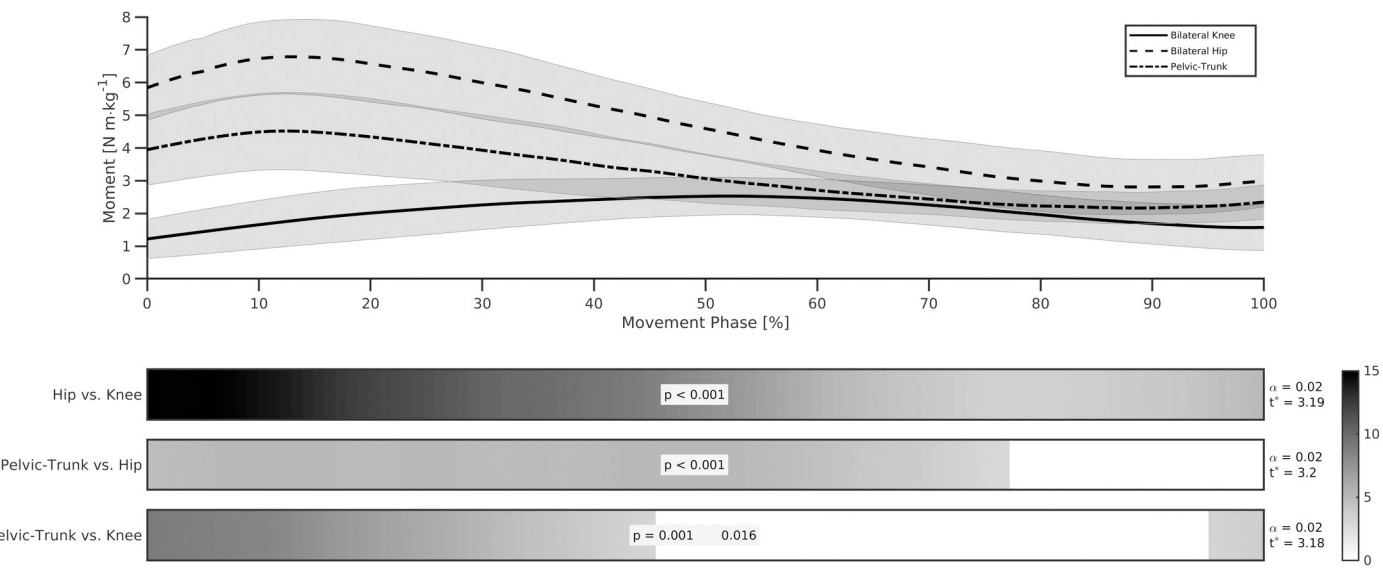

**Fig 5. Ensemble group mean (± SD gray shaded areas) joint moment time series data for the pelvic-trunk, bilateral hip and bilateral knee joints.** Shaded bars represent the SnPM{t} output statistic for each comparison. Intensity of shaded areas indicate the extent to which the critical threshold (t*) was exceeded during the movement phase with a p-value given for each supra-threshold cluster superimposed on top.

offer scientists, coaches and athletes novel information to inform training practice. The results of this study demonstrated that the hip thrust places a large and significantly greater extensor demand bilaterally on the hip joint compared to the knee and pelvic-trunk joints, allowing the first hypothesis to be accepted. However, against contemporary belief [4, 5, 7, 10], the magnitude of hip extensor moment was not constant during the lift (Fig 4), but instead diminished throughout the hip extension range of motion and reached a local minimum as the hip joint approached full extension. The second study hypothesis was therefore rejected.

A unique aspect of the current study was the novel quantification of joint kinematic and kinetic data in order to explore the musculoskeletal demand placed on the lower limb and pelvic-trunk joints during the lifting phase of the hip thrust. Interestingly, joint kinetics at the ankle joint were small in magnitude and inconsistent in the direction of plantar- and dorsiflexion across the nineteen participants (Fig 4). Inter-individual differences in the direction of the ankle joint moment were likely explained through subtle changes in the center of pressure location relative to the ankle joint, potentially influenced by initial foot position that was not controlled in this study as participants were encouraged to use their habitual lifting technique. The magnitude of ankle joint moment observed for the hip thrust is contrasting to standing exercises such as the back squat [16, 22], and deadlift [19] which have been shown to elicit plantarflexion moments in excess of 1 N·m·kg$^{-1}$. The reduced demand to maintain upright posture through the foot-ground interaction due to the third point of contact (thorax and bench) may explain the low magnitude of joint kinetics observed at the ankle in the current study. This may offer an excellent practical solution for loading the hip (and to a lesser extent the knee) extensors when low ankle loading is desirable, for example during rehabilitation.

In contrast to the ankle joint, dominant extensor moments were observed at the knee, hip and pelvic-trunk joints throughout (Fig 4). Unsurprisingly given the nature of the exercise, magnitudes of extensor moment properties were significantly greater bilaterally at the hip joint compared with the knee and trunk-pelvic joints, whilst significantly greater magnitudes were also observed at the pelvic-trunk joint compared with the knee (Tables 2 and 3, Fig 5). Although the knee joint elicited the smallest magnitudes of extensor moment and impulse in

the current study, average magnitudes of unilateral peak extensor moment (1.34 ± 0.35 N m·kg$^{-1}$, Table 2) were comparable with those previously observed during the straight- (0.80 N m·kg$^{-1}$) and hexagonal- (1.46 N m·kg$^{-1}$) bar deadlift against the same 70% 1RM external load [19, 20]. Therefore, whilst the hip thrust preferentially loads the hip extensors (as expected), there is also considerable demand placed on the knee extensors during the lift. This demand is supported by previous evidence demonstrating high levels of quadriceps muscle activation during the hip thrust exercise [7], and may indicate a requirement to stabilise the knee joint through a relatively small range of motion (20 ± 7˚, Table 2).

Previous research into the hip thrust exercise has consistently speculated that there is a consistent hip extensor moment throughout hip extension [9, 10] or maximal "tension" at the hip joint at full extension [4, 5, 7], leading to the second study hypothesis. The current investigation challenged these notions for the first time by empirically demonstrating that the magnitude of hip extension moment decreased as the hip joint extended throughout the lift, reaching a local minimum near full extension (Fig 4) and permitting the second study hypothesis to be rejected. Peak unilateral hip extensor moment (3.52 ± 0.57 N m·kg$^{-1}$) during the hip thrust occurred near the onset of the lifting phase (14.3 ± 3.1%) at a joint angle of 83 ± 16˚ of hip flexion (Table 2). At the end of the lifting phase, the hip extension moment had decreased by approximately two-thirds (Fig 4). Whilst joint moments do not directly indicate tension in the muscle fibers, the current results offer biomechanical depth beyond EMG analyses to suggest that the extensor demand placed on the hip extensors is not maximised at the end range of motion in the hip thrust exercise.

Whilst the hip joint moment was not found to be consistent, it clearly remained extensor dominant throughout the entire lifting phase of the repetition (Figs 4 and 5). The maintenance of an extensor moment nearing full hip extension is in contrast with more traditional standing exercises such as the back squat, where hip extensor moments have been shown to be zero near the end range of motion [22]. Thus, the hip thrust may offer a mechanical advantage over traditional standing exercises such as the back squat or deadlift for loading the hip extensor musculature at joint angles closer to full extension, and offers some support for previous research suggesting the superiority of the hip thrust exercise based on increased gluteal activation characteristics [5–8]. Further research should seek to directly compare hip joint kinetics between the barbell hip thrust and more traditional strength training exercises in order to further understand the potential benefits of the hip thrust exercise for developing the hip extensor musculature.

Providing kinetic insight to the pelvic-trunk joint was a further novel aspect of the current study, which demonstrated an almost identical pattern of extensor moment compared with the bilateral hip joint but with a lower magnitude across the first 80% of lift time (Fig 5). The range of pelvic-trunk joint motion was small (12 ± 21˚) and the extensor work done was more than 6-fold smaller than the hip joint (Table 2), indicating that the extensor moment at the pelvic-trunk joint was predominantly acting to resist pelvic-trunk flexion. The magnitude of peak pelvic-trunk extensor moment (4.93 ± 1.53 N m·kg$^{-1}$) was comparable with those reported during back squat and deadlift variations at similar relative loading conditions [19, 20]. However, whilst standing exercises such as the back squat will elicit both compressive and shear forces across the lumbar spine [22], the non-axial loading nature of the hip thrust may offer a means of training the hip extensor musculature with reduced compressive forces at the lumbar spine, although further investigation is required. Again, the need for future studies comparing kinetic data between the hip thrust and other commonly prescribed training exercises is required to more empirically inform exercise selection processes [16].

The novel biomechanical analysis performed in the current study was permitted by a custom-built force plate rig, specifically designed to measure external force between the thorax

and bench, whilst traditional floor-mounted platforms measured external force between the feet and ground. Results demonstrated that the temporal pattern of resultant external force applied to the floor (feet) and bench (thorax) was similar (Fig 3), although a significantly greater proportion of the external force was applied to the floor (Table 1). From an external kinetics perspective, one proposed benefit of the hip thrust exercise is the horizontal orientation of the resultant ground reaction force vector relative to the athlete, which is proposed to elicit greater transference to athletic performances that rely on horizontal force production, such as accelerative sprinting, based on the "force-vector hypothesis" [4, 10, 12]. However, during tasks that require horizontal orientation of the ground reaction force vector relative to the global (real world) coordinate system (e.g. sprinting, jumping), the athlete typically adopts a forward leaning position, producing closer alignment between orientations of the athlete and resultant force vector in the athlete's local coordinate system [11]. Therefore, when considered in the athlete's local coordinate system, there may be greater disparity between the hip thrust compared with traditional standing exercises such as the back squat or deadlift. Future quantification of the orientation of the external force vector from the floor relative to the orientation of the pelvis could offer further insight to the "force-vector hypothesis" as well as delineating differences between horizontal (hip thrust) and vertical (squat and deadlift) loaded training exercises.

Previous investigations have suggested the hip thrust to be a superior training method for enhancing horizontally oriented performances (e.g. sprinting) [8, 12], although evidence from intervention studies remains inconclusive [10, 11, 13]. Novel biomechanical evidence from this study indicates a large potential for the hip thrust to develop force-producing capabilities of the hip extensor musculature, supporting its use within strength and conditioning practice. However, the current study provides mechanistic insight to explain equivocal evidence of hip thrust superiority, as hip extensor moments were not found to be consistent throughout the lift, which is often cited as a mechanism by which the hip thrust may offer superior benefit over traditional vertical-based resistance exercises [10]. The current study did not undertake EMG analysis, which can offer insight to the muscles contributing to the observed net joint kinetics [30]. From a hip extensor perspective, research has indicated dominant EMG activity in the gluteus maximus during the hip thrust [9] but biceps femoris during accelerative [31] and maximal velocity sprinting [32]. The different contribution of each muscle group toward hip extension between these tasks may offer additional challenge to the force-vector hypothesis, and offers an insightful avenue for future research.

## Conclusions

The barbell hip thrust places extensor demand on the knee, hip and pelvic-trunk joints, with the largest bilateral demand placed on the hip extensor musculature. Whilst the hip thrust elicits a hip extensor moment throughout the full range of joint extension, the moment is not consistent and declines from the initial flexed position to full extension during the lifting phase. Whilst the current study has offered a unique dataset and novel biomechanical insight to the hip thrust exercise that offers practitioners with new information to guide exercise selection, future research is required to understand how hip thrust kinetics compare with more traditional training exercises, to understand the mechanisms through which this exercise may be a preferential training method for enhancing athletic performance.

### Practical implications

The large demand placed on the hip extensor musculature during the hip thrust exercise observed in the current study supports the hip thrust exercise as a training means for

developing the hip extensor musculature. The occurrence of a hip extensor moment as the hip joint approaches full extension is one mechanism through which the hip thrust may benefit tasks that require high hip extension force when the hip is near full extension, although further investigation is required.

## Supporting information

**S1 Appendix. Video file demonstrating example hip thrust trial in Visual 3D.**
(MP4)

## Acknowledgments

The authors would like to thank Mike Long, Ben Robson, Adam Tossell, Morgan Jones, Clémence Pouget and Jordan Lewis for their assistance with data collection and initial data processing.

## Author Contributions

**Conceptualization:** Adam Brazil, Laurie Needham, Jac L. Palmer, Ian N. Bezodis.

**Data curation:** Adam Brazil, Laurie Needham.

**Formal analysis:** Adam Brazil, Laurie Needham, Jac L. Palmer, Ian N. Bezodis.

**Investigation:** Adam Brazil, Laurie Needham, Jac L. Palmer.

**Methodology:** Adam Brazil, Laurie Needham, Jac L. Palmer, Ian N. Bezodis.

**Project administration:** Ian N. Bezodis.

**Supervision:** Ian N. Bezodis.

**Writing – original draft:** Adam Brazil.

**Writing – review & editing:** Adam Brazil, Laurie Needham, Jac L. Palmer, Ian N. Bezodis.

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
