## [Decision Letter · Decision Letter 0]

10 Dec 2020

PONE-D-20-27535

A comprehensive biomechanical analysis of the barbell hip thrust

PLOS ONE

Dear Dr. Brazil,

Thank you for submitting your manuscript to PLOS ONE. After careful consideration, we feel that it has merit but does not fully meet PLOS ONE’s publication criteria as it currently stands. Therefore, we invite you to submit a revised version of the manuscript that addresses the points raised during the review process.

We look forward to receiving your revised manuscript.

Kind regards,

Matti Douglas Allen, PhD

Academic Editor

PLOS ONE

Journal Requirements:

Reviewers' comments:

Reviewer's Responses to Questions

**Comments to the Author**

1. Is the manuscript technically sound, and do the data support the conclusions?

Reviewer #1: Yes

Reviewer #2: Yes

2. Has the statistical analysis been performed appropriately and rigorously? 

Reviewer #1: Yes

Reviewer #2: Yes

3. Have the authors made all data underlying the findings in their manuscript fully available?

Reviewer #1: Yes

Reviewer #2: Yes

4. Is the manuscript presented in an intelligible fashion and written in standard English?

Reviewer #1: Yes

Reviewer #2: Yes

5. Review Comments to the Author

Reviewer #1: General comments

The focus of the study is the biomechanics of the barbell hip thrust exercise and a research article that contributes to practice. However, there are a few general comments to be completed. Firstly, a free body diagram might be added to the materials and methods section. In addition, a figure can be used showing the experimental setup. Secondly, Newton-Euler inverse dynamic formulas might be presented in the method section.

Detailed comments about this study are below.

Abstract

The results of the study are presented clearly in abstract section.

Introduction

The reasons for studying the biomechanics of hip thrust exercise are presented in the introduction section. However, the hypothesis of the study has been not included in the introduction section. What is the hypothesis of the study? Please add hypothesis to the introduction.

Methods

Please, add a free body diagram to the materials and methods section. A considerable simplification can be made to understand the kinetic and kinematic dimensions of the hip thrust movement if the forces and moments in the pelvic-trunk joint and lower extremity joints are defined by the free body diagram. In addition, a figure that shows the marker set attached to the body can be added to method section, or a figure showing the experimental setup. As known, Euler’s 3D equations of motion are used for kinetic analyses of a segment Please, add to the text Newton-Euler inverse dynamic procedures used for calculation of moment. On the other hand, Euler's equations also require data such as angular acceleration and moments of inertia of lower extremity segments (thigh, shank and foot). It can be mentioned in the text if angular acceleration and moments of inertia of the lower limb segments data were used in calculation.

Results

External kinematics and kinetics and joint kinematics and kinetics were presented clearly in Table1, Table2, and Table3, and were supported with the figures.

Discussion

It was reported in the study that the hip extensor moment decreased during the lift contrary to the literatüre. This is the most important result of the study. The main results of the study, which contradicts with the literature, are very well discussed with fine details. I applaud the authors.

Page 14, line 284

Both dot and comma were used together in the end of the sentence. Please, remove the comma.

Page 14, line 303

Please, add a dot at the end of the sentence.

Reviewer #2: Manuscript Number: PONE-D-20-27535

Manuscript Title: A comprehensive biomechanical analysis of the barbell hip thrust

Comments to the Author

This paper attempts to identify the kinematic and kinetic characteristics of the hips-thrust exercise. The measurement method is original and interesting. While I find this manuscript and the topic interesting and of relevance to the readership of Plos One. These findings are expected to be useful for strength training. However, there are some shortcomings in the presentation.

Introduction

Page 4, Lines60-65

Clarifying the relevance of why previous studies(10,11,13) differ in their results and the need for comprehensive biomechanical analysis will make the significance of this study clearer. I recommend the author to revise it.

Page 4, Lines 71-72

The need and importance of investigating the musculoskeletal demands on the lower extremity and pelvic-trunk joints is not only cited, but also written, which makes the significance of analyzing the lower extremity and pelvic-trunk joints of the hip-thrust exercise in this study more clear. I recommend the author to revise it.

Page 4, Lines 77-81

Having a hypothesis often makes a paper easier to read. Authors are encouraged to state their hypothesis if they can derive it from the background already described.

Methods

Page 5, Line 85

Please indicate what the numbers indicate about the characteristics of the subjects: "age," "height," and "weight.

Page 5, Line 90

Add the details of the marker positions. This is important for the reproducibility of the experiments.

Page 5, Line 90

The setting of the measurement is one of the originalities of this study. I believe that illustrations or photographs of the measurements will help the reader's understanding and reproducibility of the experiment, and will enhance the value of the paper. It also helps the reader to understand by describing what new things this original setting allows us to know.

Page 5, Lines 103

The method for measuring the 1RM of hip thrust needs to be specified. It is also necessary to specify how the start and end postures and the speed of movement were controlled during the 1RM measurements and experiments. Since the motion and load conditions are not independent variables in this study, a description of the control of the motion is essential.

Page 5, Line 107

Although relatively slow motion is the target of the analysis, I feel that the cutoff frequency is low. I request the authors to add the evidence that the cutoff frequency of 3 Hz was appropriate for signal processing.

Page 6, Line115

With respect to the method of analysis, a description or literature on how to specifically define the local coordinate system of each segment is needed. Without it, the reader does not know the order of rotation of XYZ.

Page 6, 120

What model of inertia parameters was used in the inverse dynamics analysis? Or did you estimate inertia parameters based on morphometric measurements for each subject? Please clarify these.

Page 7 , Line114

Table 2 shows the negative d-value. I request the authors to maintain consistency with the description of the method.

Results

Page 9, Lines 177-179 & Page 13, Lines 274-276

Since Fig. 2 is an averaged figure, no information about individual differences is available in the figure. Request the authors to revise the figure or text.

Table 2 shows the negative d-value. I request the authors to maintain consistency with the description of the method.

Discussion

Pages 12-13, Lines 252-271

This section seems a bit redundant as a preamble to the results-based discussion. I recommend integrating it with the text at the end of the Discussion.

Page 13, Lines 276-279

As pointed out in the methods section, the presence or absence of movement control is one of the major factors influencing the results in the study of training movements. I request the authors to revise this text based on the revisions in the methods section.

Page 15, Lines 312-313

How much load is required for training depends on the objective. In addition, since this study has only one loading condition, what this sentence suggests seems to be an expanded interpretation of the results. Request the author to correct or delete the text.

Page 16, Liens 335-337

Future research will be aided by results-based conjecture about the possible risks of hip thrust versus squat, etc.

Conclusions

Page 17, Liens 362

What kind of movement selection can be shown in this study that has not been shown in the previous studies? (This study revealed changes in various parameters between joints and with time within the same movement. However, it does not show the superiority of hip thrust over other exercises)

Table 1

BW is not a proper unit of measurement. Please add a Note to the table or correct the units.

Table 2

No need to specify "±" in Table 2 where the results are not shown.

Fig 1

The legend in the figure does not match the title of the figure, so unification helps the reader to understand it.

Fig 2

It is recommended to add stick pictures to the figure to relate the data to the posture.

Add a horizontal line to the vertical axis of 0 to make it easier to see when the transition between flexion and extension is made.

Fig 3

I may be mistaken due to the low resolution of the figure, but what does 0.016 within Pelvic-Trunk vs Knee indicate? Please add explanations if necessary.

6. PLOS authors have the option to publish the peer review history of their article (what does this mean?). If published, this will include your full peer review and any attached files.

Reviewer #1: No

Reviewer #2: No

---

## [Author Response · Author response to Decision Letter 0]

21 Jan 2021

Editor Comments to the Author: 

2. Please include captions for your Supporting Information files at the end of your manuscript, and update any in-text citations to match accordingly. 

Response: Thank you for highlighting this. We have revised the manuscript, file names, and supporting information to meet PLOS ONES's style requirements.

Reviewer(s)’ Comments to the Author:

Reviewer 1

Comments to the Author

General Comments

The focus of the study is the biomechanics of the barbell hip thrust exercise and a research article that contributes to practice. However, there are a few general comments to be completed. Firstly, a free body diagram might be added to the materials and methods section. In addition, a figure can be used showing the experimental setup. Secondly, Newton-Euler inverse dynamic formulas might be presented in the method section.

On behalf of the authors I would like to thank you for your positive and helpful comments, which have contributed to the improvement of our manuscript. All comments have been carefully considered and where appropriate, amendments to the manuscript have been made. In particular, new Figures have been included to show the experimental setup (Figure 1) and a free body diagram (Figure 2). 

Specific comments

Abstract 

The results of the study are presented clearly in abstract section.

Response: Thank you for your positive comments.

Introduction

The reasons for studying the biomechanics of hip thrust exercise are presented in the introduction section. However, the hypothesis of the study has been not included in the introduction section. What is the hypothesis of the study? Please add hypothesis to the introduction.

Response: Thank you for raising this, along with Reviewer 2. In the revised manuscript we have included hypotheses at the end of the introduction (line 86-88) and have altered the discussion narrative to reflect the addition of hypotheses

Methods 

Please, add a free body diagram to the materials and methods section. A considerable simplification can be made to understand the kinetic and kinematic dimensions of the hip thrust movement if the forces and moments in the pelvic-trunk joint and lower extremity joints are defined by the free body diagram. In addition, a figure that shows the marker set attached to the body can be added to method section, or a figure showing the experimental setup. As known, Euler’s 3D equations of motion are used for kinetic analyses of a segment Please, add to the text Newton-Euler inverse dynamic procedures used for calculation of moment. On the other hand, Euler's equations also require data such as angular acceleration and moments of inertia of lower extremity segments (thigh, shank and foot). It can be mentioned in the text if angular acceleration and moments of inertia of the lower limb segments data were used in calculation.

Response: We agree that including visual information in regard to the experimental setup and free-body diagram would benefit the manuscripts reach and readability. Therefore, new Figures have been included to show the experimental setup (Figure 1) and a free body diagram (Figure 2). The marker-set has been detailed in lines (102-106). 

With respect to inverse dynamics procedures, we have revised the sentence structure in lines 148-151, to reflect and reference the use of standard procedures within Visual 3D software. The methods used for calculating joint kinetic data is publicly available on the Visual 3D website and based on the methods cited in ref [24]. The authors feel it is not necessary to also add equations to the manuscript. 

Results

External kinematics and kinetics and joint kinematics and kinetics were presented clearly in Table1, Table2, and Table3, and were supported with the figures.

Response: Thank you for your positive comments on the results section. The results section has undergone minor amendments as per the recommendations of Reviewer 2 (please see below for details). 

Discussion

It was reported in the study that the hip extensor moment decreased during the lift contrary to the literature. This is the most important result of the study. The main results of the study, which contradicts with the literature, are very well discussed with fine details. I applaud the authors.

Response: Again, thank you for your very positive comments. We have addressed your points below, and made further revisions based on the recommendations of Reviewer 2 (please see below for details). 

• Page 14 Line 284. Both dot and comma were used together in the end of the sentence. Please, remove the comma.

Response: Amended in revised manuscript.

• Page 14 Line 303. Please, add a dot at the end of the sentence.

Response: Amended in revised manuscript.

Reviewer 2

Comments to the Author

General Comments

This paper attempts to identify the kinematic and kinetic characteristics of the hips-thrust exercise. The measurement method is original and interesting. While I find this manuscript and the topic interesting and of relevance to the readership of Plos One. These findings are expected to be useful for strength training. However, there are some shortcomings in the presentation.

On behalf of the authors I would like to thank you for your helpful and positive comments, that has contributed to the improvement of our manuscript. All of your comments have been carefully considered, and where appropriate amendments to the manuscript have been made.

Specific Comments

Introduction

• Page 4, Line 60-65. Clarifying the relevance of why previous studies (10,11,13) differ in their results and the need for comprehensive biomechanical analysis will make the significance of this study clearer. I recommend the author to revise it.

Response: We have revised the structure of the manuscript around this sentence to help identify the significance of the study (line 71-76), before the original narrative that builds rationale for the biomechanical analysis conducted (line 77-84).

• Page 4, Line 71-72. The need and importance of investigating the musculoskeletal demands on the lower extremity and pelvic-trunk joints is not only cited, but also written, which makes the significance of analyzing the lower extremity and pelvic-trunk joints of the hip-thrust exercise in this study more clear. I recommend the author to revise it..

Response: We are not sure specifically what the reviewer is asking in this comment. The use of references 17-22 is to highlight that a breadth of research has been dedicated to performing joint kinetic analysis of strength training exercises, and that this effective approach has yet to be performed for the hip thrust. We believe this provides a strong rationale for the current study. 

• Page 4, Line 77-81. Having a hypothesis often makes a paper easier to read. Authors are encouraged to state their hypothesis if they can derive it from the background already described.

Response: Thank you for raising this, along with Reviewer 1. In the revised manuscript we have included hypotheses at the end of the introduction (line 86-88) and have altered the discussion narrative to reflect the addition of hypotheses: 

Methods

• Page 5, Line 85. Please indicate what the numbers indicate about the characteristics of the subjects: "age," "height," and "weight.

Response: This has been amended in the revised manuscript (line 94-95).

• Page 5, Line 90. Add the details of the marker positions. This is important for the reproducibility of the experiments.

Response: The revised manuscript now contains a list of maker positions used (line 102-106).

• Page 5, Line 90. The setting of the measurement is one of the originalities of this study. I believe that illustrations or photographs of the measurements will help the reader's understanding and reproducibility of the experiment and will enhance the value of the paper. It also helps the reader to understand by describing what new things this original setting allows us to know.

Response: Thank you for this comment. In line with comments from both reviewers, the revised manuscript contains two new figures in the methods section. One demonstrates the experimental setup (Figure 1), and the other a free-body diagram of the hip thrust exercise (Figure 2). We have also added new supplementary material alongside the manuscript, which shows a Visual 3D video of an example hip thrust trial. 

• Page 5, Line 103. The method for measuring the 1RM of hip thrust needs to be specified. It is also necessary to specify how the start and end postures and the speed of movement were controlled during the 1RM measurements and experiments. Since the motion and load conditions are not independent variables in this study, a description of the control of the motion is essential.

Response: As the participants were well-trained individuals regularly performing the hip thrust exercise, 1RM was self-reported by the participants based on their most recent training performances. To maintain ecological validity, participants were asked to execute the movement using their habitual technique and timing, with the instruction to perform each repetition with maximum intent during the lifting phase to full hip extension, before controlling the barbell back to the starting position. Each repetition was initiated from this starting position, with the barbell coming to a rest on the ground between each of the three repetitions. This has been detailed in the revised manuscript (lines 119-124). 

• Page 5, Line 107. Although relatively slow motion is the target of the analysis, I feel that the cut-off frequency is low. I request the authors to add the evidence that the cut-off frequency of 3 Hz was appropriate for signal processing.

Response: Thank you for your comment. The cut-off frequency used in the filtering process was based on residual analysis techniques (Winter, 2009). This statement and reference has been added to the revised manuscript (line 133-134). Similar studies in the area (below) have utilised cut-off frequencies in the 3-6 Hz range for kinematic data, aligning with the current study. Further, the frequency of the movement being analysed is well below 3 Hz, further justifying this as the cut-off frequency

Swinton PA, Lloyd R, Keogh JW, Agouris I, Stewart A. A Biomechanical Comparison of the Traditional Squat, Powerlifting Squat, and Box Squat. The Journal of Strength and Conditioning Research. 2012;26(7):1805-1816.

Bryanton, Megan A., et al. "Effect of squat depth and barbell load on relative muscular effort in squatting." The Journal of Strength & Conditioning Research 26.10 (2012): 2820-2828.

Gullett, Jonathan C., et al. "A biomechanical comparison of back and front squats in healthy trained individuals." The Journal of Strength & Conditioning Research 23.1 (2009): 284-292.

Southwell, Daniel J., et al. "The effects of squatting footwear on three-dimensional lower limb and spine kinetics." Journal of Electromyography and Kinesiology 31 (2016): 111-118.

• Page 6, Line 115. With respect to the method of analysis, a description or literature on how to specifically define the local coordinate system of each segment is needed. Without it, the reader does not know the order of rotation of XYZ.

Response: To provide clarity in the local coordinate system (SCS), the manuscript has been amended to include reference to the SCS being defined in the same right-handed orthogonal coordinate system as the global reference (line 145-146).

• Page 6, Line 120. What model of inertia parameters was used in the inverse dynamics analysis? Or did you estimate inertia parameters based on morphometric measurements for each subject? Please clarify these.

Response: The segment mass an inertial parameters used were consistent with the Visual 3D default (Dempster & Hanavan). The manuscript has been updated accordingly (line 148-151), referencing the source of the information and highlighting that all inverse dynamics procedures were completed using the default methods in Visual 3D (as with Southwell et al., 2016). 

• Page 7, Line 124. Table 2 shows the negative d-value. I request the authors to maintain consistency with the description of the method.

Response: To address both directions of ‘d’, we have included two amendments to the revised manuscript. The first, is the addition of “equivalent scale used for negative values of d” in line 181. Secondly, in the footer of Table 3, we have now added the wording “the direction of d indicates whether the joint on the left (positive) vs. right (negative) was of larger magnitude.” To give context to the direction of ‘d’ for each comparison. 

Results

• Page 9, Line 177-179/ Page 13, Line 274-276. Since Fig. 2 is an averaged figure, no information about individual differences is available in the figure. Request the authors to revise the figure or text.

Response: We feel that the point of discussion is valid based on the original construction of Figure 2 (now Figure 3) that shows the ensemble group mean and standard deviation. The shading in Figure 3 that covers both positive and negative values of ankle joint moment represents the variability between the nineteen participants in the study, and it is valid to interpret the standard deviation as inter-individual variation across the sample. Whilst the figure does not allow direct comparison of individual participants, the group standard deviations indicates that there was inter-individual variation in the nature of dorsi- / plantar-flexion moment during the lift. 

In the revised manuscript, “inter-individual” has been removed from the results narrative (line 2010-211) and figure captions have been modified to explicitly state that data shown is ensemble group mean and standard deviation.

• Table 2. Shows the negative d-value. I request the authors to maintain consistency with the description of the method.

Response: Please see response above regarding the changes made in relation to this comment. 

Discussion

• Page 12-13, Line 252-271. This section seems a bit redundant as a preamble to the results-based discussion. I recommend integrating it with the text at the end of the Discussion.

Response: Thank you for this comment. We agree that this section is better placed toward the end of the discussion and have moved the section accordingly in the revised manuscript (line 356-375).

• Page 13, Line 276-279. As pointed out in the methods section, the presence or absence of movement control is one of the major factors influencing the results in the study of training movements. I request the authors to revise this text based on the revisions in the methods section.

Response: In addition to the justification offered above, the focus of these lines was to offer mechanical explanation for the standard deviation overlapping dorsi- and plantar-flexion moments shown in Figure 2. Participants were encouraged to perform the movement with their habitual technique which likely resulted in the observed variation, and has been added to the manuscript (line 295). 

• Page 15, Line 312-313. How much load is required for training depends on the objective. In addition, since this study has only one loading condition, what this sentence suggests seems to be an expanded interpretation of the results. Request the author to correct or delete the text.

Response: The aim of the sentence is to summarise one of the key findings of the study, which is that the hip joint moment was not maximal at full extension, which has been postulated in previous studies using EMG analysis. To keep the focus on the current results, and not speculate on advised range of motion to be used in training, the final part of the sentence has been deleted. 

• Page 16, Line 335-337. Future research will be aided by results-based conjecture about the possible risks of hip thrust versus squat, etc.

Response: We feel as if the comment is addressed in the current narrative, reflecting that further investigation is required in regard to spinal loading, as well as highlighting the need for empirical studies comparing the hip thrust with other common training exercises.

Conclusions

• Page 17, Line 362. What kind of movement selection can be shown in this study that has not been shown in the previous studies? (This study revealed changes in various parameters between joints and with time within the same movement. However, it does not show the superiority of hip thrust over other exercises).

Response: Thank you for this comment. The reference to “exercise selection” in the manuscript is the process through which coaches/ athletes decide on which exercises to utilise within training. The new insight to the kinetic demands of the hip thrust exercise from the current study may offer rationale for including/ excluding the hip thrust based on its underlying mechanical features. For example, the study has highlighted that indeed the hip thrust does place a large demand on the hip extensors, but not a consistent demand throughout the movement, or at the end range of hip extension, which has been used as rationale for the exercises’ benefit through previous research. It is not the intention of the authors to suggest its superiority here, but to recognise that increased biomechanical knowledge of the exercise can inform the exercise selection process. We feel that the current wording in the manuscript reflects this explanation.

Tables & Figures

• Table 1. BW is not a proper unit of measurement. Please add a Note to the table or correct the units.

Response: A footnote to Table 1 has been added to the revised manuscript (line 200-201), detailing that external force data have been normalised to bodyweight (BW). This procedure has also been defined in the methods section (line 43) and is commonly used in sports biomechanics literature to report external force data. 

• Table 2. No need to specify "±" in Table 2 where the results are not shown.

Response: Thank you for noticing this. 

• Figure 1. The legend in the figure does not match the title of the figure, so unification helps the reader to understand it.

Response: Thank you for noticing this. Both the caption and legend for Figure 1 (now Figure 3) has been amended in the revised manuscript.

• Figure 2. It is recommended to add stick pictures to the figure to relate the data to the posture.

Add a horizontal line to the vertical axis of 0 to make it easier to see when the transition between flexion and extension is made.

Response: In the revised manuscript, a horizontal line at the y-axis 0-intercept has been added to better show the transition between positive and negative phases of moment/ power. The authors wish to exclude further information from the figure (e.g. stick pictures) to encourage the reader to consider the entire lifting phase, and not specific, discrete points when viewing this figure. In addition, stick figures are not commonly utilised in this type of analysis (e.g. Swinton et al., 2011; 2012, Southwell et al., 2016).

• Figure 3. I may be mistaken due to the low resolution of the figure, but what does 0.016 within Pelvic-Trunk vs Knee indicate? Please add explanations if necessary.

Response: The SnPM inference process returns a unique p-value for each supra-threshold cluster. The scalar output statistic (SnPm{t}) for the Pelvic-Truck vs Knee interaction exceeded the critical threshold at two unique time periods. As such the value in question represents the adjusted p-value for the second supra-threshold cluster. This has been clarified in the caption for Figure 3 (now Figure 5 in the revised manuscript).

---

## [Decision Letter · Decision Letter 1]

4 Mar 2021

PONE-D-20-27535R1

A comprehensive biomechanical analysis of the barbell hip thrust

PLOS ONE

Dear Dr. Brazil,

Thank you for submitting your manuscript to PLOS ONE. After careful consideration, we are pleased to notify you your paper will be accepted at PLOS ONE pending the very minor revision suggested by the reviewer. Therefore, we invite you to submit a revised version of the manuscript that addresses the points raised during the review process.

Once this revision is complete, we anticipate a very quick acceptance and initiation of the final stages in publication.

We look forward to receiving your revised manuscript.

Kind regards,

Matti Douglas Allen, PhD

Academic Editor

PLOS ONE

Journal Requirements:

Reviewers' comments:

Reviewer's Responses to Questions

**Comments to the Author**

1. If the authors have adequately addressed your comments raised in a previous round of review and you feel that this manuscript is now acceptable for publication, you may indicate that here to bypass the “Comments to the Author” section, enter your conflict of interest statement in the “Confidential to Editor” section, and submit your "Accept" recommendation.

Reviewer #1: All comments have been addressed

Reviewer #2: (No Response)

2. Is the manuscript technically sound, and do the data support the conclusions?

Reviewer #1: Yes

Reviewer #2: Yes

3. Has the statistical analysis been performed appropriately and rigorously? 

Reviewer #1: Yes

Reviewer #2: Yes

4. Have the authors made all data underlying the findings in their manuscript fully available?

Reviewer #1: Yes

Reviewer #2: Yes

5. Is the manuscript presented in an intelligible fashion and written in standard English?

Reviewer #1: Yes

Reviewer #2: Yes

6. Review Comments to the Author

Reviewer #1: (No Response)

Reviewer #2: Manuscript Number: PONE-D-20-27535R1

Manuscript Title: A comprehensive biomechanical analysis of the barbell hip thrust

Comments to the Author

Thanks to authors for making appropriate revisions. I believe that this research will develop resistance training and help athletes and coaches.

Mayor comments;

Figure 2

How did you handle the moment acting between the toe and the ground (I guess authors assumed that moment to be zero)? The details of the method should be clearly stated.

7. PLOS authors have the option to publish the peer review history of their article (what does this mean?). If published, this will include your full peer review and any attached files.

Reviewer #1: No

Reviewer #2: No

---

## [Author Response · Author response to Decision Letter 1]

5 Mar 2021

Journal Requirements.

Please review your reference list to ensure that it is complete and correct. If you have cited papers that have been retracted, please include the rationale for doing so in the manuscript text or remove these references and replace them with relevant current references. Any changes to the reference list should be mentioned in the rebuttal letter that accompanies your revised manuscript. If you need to cite a retracted article, indicate the article’s retracted status in the References list and also include a citation and full reference for the retraction notice.

Thank you for this comment. From the original submission there have been no references removed from the manuscript. Following the first revision, three references were added to the manuscript (23, 25, 26) to address reviewer comments, which resulted in changes to the numerical order of the in-text citations. The first revision of the manuscript highlighted the changes to the numerical order of in-text citations, but this did not reflect a removal/ change in references used. Since the original submission, reference 8 had been allocated a volume and page number, which was updated in the first revision of the manuscript.

Reviewer(s)’ Comments to the Author:

Specific comments

Figure 2. How did you handle the moment acting between the toe and the ground (I guess authors assumed that moment to be zero)? The details of the method should be clearly stated.

• In the current analysis, the foot was modelled as a single rigid segment with the distal endpoint at the metatarsophalangeal joint (MTPJ). The moment between the distal end of the foot and ground was not included in our analysis and was assumed to be zero. As the foot was modelled as a single segment, the joint moment acting at the MTPJ (between foot and toe) was also not considered in our analysis, which is consistent in the resistance training literature. Whilst the MTPJ has been shown to generate and absorb energy in locomotive tasks (e.g. sprinting), evidence has suggested (e.g. Bezodis et al., 2012. DOI: https://doi.org/10.1123/jab.28.2.222) that if the specific focus of the study is the kinetics of the ankle, knee and hip, the current method is appropriate.

In the revised manuscript, the following sentence has been included to the methods section for clarity (line 151-154):

“The distal end of the foot was defined at the metatarsophalangeal joint, and the moment acting between the distal end of the foot and the ground was assumed to be zero.”

---

## [Editor Report · Decision Letter 2]

16 Mar 2021

A comprehensive biomechanical analysis of the barbell hip thrust

PONE-D-20-27535R2

Dear Dr. Brazil,

We’re pleased to inform you that your manuscript has been judged scientifically suitable for publication and will be formally accepted for publication once it meets all outstanding technical requirements.

Kind regards,

Matti Douglas Allen, MD, PhD

Academic Editor

PLOS ONE

---

## [Editor Report · Acceptance letter]

19 Mar 2021

PONE-D-20-27535R2 

A comprehensive biomechanical analysis of the barbell hip thrust 

Dear Dr. Brazil:

I'm pleased to inform you that your manuscript has been deemed suitable for publication in PLOS ONE. Congratulations! Your manuscript is now with our production department. 

Kind regards, 

on behalf of

Dr. Matti Douglas Allen 

Academic Editor

PLOS ONE